# Influence of Heating during Cooking on *Trans* Fatty Acid Content of Edible Oils: A Systematic Review and Meta-Analysis

**DOI:** 10.3390/nu14071489

**Published:** 2022-04-02

**Authors:** Saiuj Bhat, Damian Maganja, Liping Huang, Jason H. Y. Wu, Matti Marklund

**Affiliations:** 1Department of Vascular Surgery, Royal Perth Hospital, Victoria Square, Perth, WA 6000, Australia; saiujbhat@gmail.com; 2The George Institute for Global Health, University of New South Wales, Sydney, NSW 2050, Australia; dmaganja@georgeinstitute.org.au (D.M.); hliping@georgeinstitute.org.au (L.H.); jwu1@georgeinstitute.org.au (J.H.Y.W.); 3Department of Epidemiology, Johns Hopkins Bloomberg School of Public Health, Baltimore, MD 21205, USA; 4Department of Public Health and Caring Sciences, Uppsala University, SE75105 Uppsala, Sweden

**Keywords:** cooking, frying, baking, food policy

## Abstract

Consumption of *trans* fatty acids (TFA) is associated with adverse health outcomes and is a considerable burden on morbidity and mortality globally. TFA may be generated by common cooking practices and hence contribute to daily dietary intake. We performed a systematic review and meta-analysis to investigate the relationship between heating edible oils and change in their TFA content. A systematic search of experimental studies investigating the effect of various methods of heating on TFA content of edible oils was conducted in Medline and Embase since their inception up to 1 October 2020 without language restrictions. Comparable data were analysed using mixed multilevel linear models taking into account individual study variation. Thirty-three studies encompassing twenty-one different oils were included in this review. Overall, heating to temperatures <200 °C had no appreciable impact on different TFA levels. Between 200 and 240 °C, levels of C18:2 t (0.05% increase per 10 °C rise in temperature, 95% CI: 0.02 to 0.05%), C18:3t (0.18%, 95% CI: 0.14 to 0.21%), and total TFA (0.38%, 95% CI: 0.20 to 0.55%) increased with temperature. A further increase in total TFA was observed with prolonged heating between 200 and 240 °C. Our findings suggest that heating edible oils to common cooking temperatures (≤200 °C) has minimal effect on TFA generation whereas heating to higher temperatures can increase TFA level. This provides further evidence in favour of public health advice that heating oils to very high temperatures and prolonged heating of oils should be avoided.

## 1. Introduction

*Trans*-fatty acids (TFA) are fatty acids with at least one unsaturated, non-conjugated double bond in the *trans* configuration. Consumption of TFA is associated with cardiovascular disease risk factors including increased low-density lipoprotein cholesterol, triglyceride, and lipoprotein (a) levels, decreased high-density lipoprotein cholesterol, and increased insulin resistance, adiposity, and endothelial dysfunction [1]. Increased TFA consumption has also been associated with greater incidence of diabetes and coronary heart disease [2,3,4]. Approximately 645,000 annual deaths globally are attributed to a diet high in TFA [5]. In view of its adverse health effects, the World Health Organisation recommends that TFA intake be minimised and contribute no more than 1% of total daily energy intake [6], in line with other food and health advisory organisations [7]. While global TFA intake appears to have gradually declined over the past three decades [8], consumption of TFA remains high in many parts of the world, accounting for up to 6.5% of total daily energy [9].

*Trans* fatty acids in diet originate from naturally occurring TFA from ruminants or industrially produced TFA formed by partial hydrogenation of vegetable oils. In addition, thermal treatment of cooking oils can alter their physicochemical properties and potentially generate TFA from cis-unsaturated fatty acids during cooking procedures (e.g., deep-, pan-, or stir-frying) [10,11,12]. While some previous studies have reported increases in TFA content of edible oils during cooking, others demonstrate little or no change [13,14,15]. Furthermore, amongst studies that report increases in TFA content of oil during cooking, the magnitude of the reported increase is variable. No systematic review has assessed the impact of heating on TFA content in cooking oils. Therefore, the aim of our study was to perform a systematic review of experimental studies to determine how thermal treatment influences the TFA content of edible oils. We hypothesised that an increase in heating temperature and time would be associated with an increase in the TFA content of edible oils, irrespective of the method of heating.

## 2. Materials and Methods

This systematic review followed the Preferred Reporting Items for Systematic Reviews and Meta-Analyses (PRISMA) guidelines and was registered on PROSPERO [CRD42020223609]. Two reviewers independently screened studies for eligibility (LH, SB, or DM) and extracted data (SB, DM). Discrepancies were resolved by consensus or involvement of a third reviewer (MM).

### 2.1. Search Strategy

A systematic literature search, up to 1 October 2020, was conducted in Medline and Embase without date or language restrictions. Full search terms are available in Appendix B. Reference lists of eligible studies were manually scanned to identify additional relevant publications.

### 2.2. Eligibility Criteria

Experimental studies quantifying the TFA content of edible, commercially available cooking oils before and after various common cooking practices, including heating, deep-frying, stir-frying, pan-frying, and baking, were included. Studies that did not measure changes in TFA composition of oils were excluded.

### 2.3. Data Extraction

The following data were extracted from included studies: publication information, type of oil, type of foods (if any) used in the experiment, type of TFA studied, experimental conditions including cooking method, temperature, heating time, heating cycle, and heating vessel, method of quantifying fatty acid composition, TFA level and units of measure, estimates of uncertainty, and number of replicates. Cooking oils that contained mixtures of different oils (e.g., sunflower, safflower, and canola oils) were consolidated into a category labelled “blend”; vanaspati, bakery shortening, and other hydrogenated oils were consolidated into the “hydrogenated vegetable fat” category.

### 2.4. Risk of Bias Assessment

Included studies were evaluated against the following criteria to determine study quality: reporting of analytical variability (i.e., reporting of coefficient of variation) (yes or no), more than one replicate (yes or no), and reporting of measures of uncertainty (SD, SE, or 95% CI) (yes or no). Studies scored a point for each of the criteria fulfilled. Studies that did not report measures of uncertainty for some outcomes received a score of 0 for that criterion.

### 2.5. Data Analysis

The primary outcome was change in level of TFA (% of total fatty acids), calculated as the mean difference between TFA levels before and after the experiment. Comparable data from included studies were analysed using mixed multilevel linear models, with intercepts for individual studies specified as random effects to account for variation in experimental conditions and oils between studies. Differences in mean TFA levels between predefined heating temperature intervals were assessed in models with temperature coded as an ordinal variable (room temperature [control], <200 °C, 200–240 °C, and >240 °C). The temperature intervals were defined based on usual cooking temperatures (i.e., common cooking methods rarely utilise temperatures above 200 °C) and smoke points of commonly used cooking oils (i.e., the smoke points of most commonly used cooking oils do not exceed 240 °C). To assess linear associations of heating temperature and TFA content within temperature intervals, we utilised linear splines with knots at 200 °C and 240 °C. Two models were evaluated to assess the impact of cooking temperature on TFA content: (1) crude, without adjustment; and (2) adjusted for heating time and type of oil used. Subgroup analyses for each type of oil were not conducted given the small number of data points available for individual cooking oils. We investigated the interaction between heating temperature and heating time for those temperature intervals where a significant change in TFA levels was observed. All statistical tests were performed using STATA 16 (Stata Corp, College Station, TX, USA), with two-tailed alpha of 0.05. Data are presented as mean ± standard deviation (SD) or median [interquartile range, IQR] unless stated otherwise.

## 3. Results

### 3.1. Study Characteristics

Of the 234 studies identified by our search, 33 were included in this review (Figure 1 and Appendix A). Collectively, the studies analysed 21 different cooking oils, with corn [11,13,16,17,18,19,20,21,22,23], soybean [15,20,21,23,24,25,26,27,28,29], sunflower [11,16,20,21,30,31,32,33,34], and hydrogenated vegetable fat [15,24,27,29,31,35,36,37] being the most commonly assessed; fewer studies investigated other oils including Aleppo pine seed [38], blend [16,17,23,33,39,40], canola [17,41], coconut [42], cottonseed [35], groundnut [24], linseed [20], olive [11,12,16,20,24,32,34], palm [33,36,39,42,43], peanut [20,28,36], peony seed [20], rapeseed [20,24,44], rice bran [17,20,40], safflower [17,23], and sesame [17,20] oil; or solid fats such as ghee [24] and lard [16]. The relatively high baseline (preheating) levels of TFA in hydrogenated vegetable fats compared to the two most commonly studied cooking oils (corn and soybean) are shown in Appendix A. While all studies evaluated the change in TFA content of cooking oils, only seven investigated the change in TFA content of foods—chicken, fish, or potato—cooked in various oils [18,21,27,32,35,43,45]. Quality assessment of included studies is shown in Appendix A. Only 3 studies (9%) were deemed to have the highest quality score (3/3) in our assessment, 15 (45%) scored 2 out of 3, 9 studies (27%) scored 1 out of 3, and 6 studies (18%) scored 0. The major quality limitation was lack of reporting of analytic variability across the included studies.

Trans fatty acids investigated included *trans*-palmitoleic acid (C16:1t), elaidic acid (C18:1t), *trans*-linoleic acid (C18:2t), *trans*-linolenic acid (C18:3t), *trans*-eicosenoic acid (C20:1t), monounsaturated TFA, polyunsaturated TFA, and total TFA. The median TFA level at baseline was 0.23% of total fatty acids [0.04% to 0.70%]. Five different cooking methods were employed: baking [13], heating [19,24,30,33,36,38], deep-frying [11,13,17,18,20,21,23,24,27,28,29,32,33,34,35,37,39,40,41,42,43], pan-frying [11,13,20], and stir-frying [13,20]. Heating temperatures, heating times, and heating cycles corresponding to the various cooking methods and for the various TFA studied are summarised in Appendix A. Across studies and experiments, the median heating temperature was 180 °C [IQR: 175–200 °C], the median heating time was 45 min [IQR: 6–480 min], and the median cooking cycle was 1 [IQR: 1–2 cycles].

### 3.2. Heating Temperature and Change in Level of TFA

The median TFA levels (shown separately for subtypes of TFA and total TFA) at different temperature intervals are depicted in Table 1. Compared to room temperature, heating oils to temperatures <200 °C (i.e., common cooking temperatures) had no significant impact on TFA levels (Figure 2 and Appendix A). However, within this temperature range, C18:3t levels increased significantly, albeit modestly, with higher temperature (0.02% increase per 10 °C rise in temperature, 95% CI: 0.01 to 0.02%). While mean TFA levels in oils heated to between 200 and 240 °C did not differ significantly from TFA levels in unheated or less heated oils, the levels of C18:2t (0.05% increase per 10 °C rise in temperature, 95% CI: 0.02 to 0.05%), C18:3t (0.18%, 95% CI: 0.14 to 0.21%), and total TFA (0.38%, 0.20 to 0.55%) increased with temperature within this temperature range.

As expected, the change in TFA was more striking, with an increase in heating time at temperatures between 200 and 240 °C (Appendix A). For example, when heated for 6 h, total TFA levels increased by 0.86% for every 10 °C rise in heating temperature between 200 and 240 °C compared to no significant change after 15 or 45 min of heating (Figure 3).

Although very few (*n* = 3) studies evaluated temperatures above 240 °C, mean levels of individual TFA (but not total TFA) in oils heated to these temperatures were overall significantly greater compared to oils heated to <200 °C. Within this temperature range, C18:1t (0.72% increase per 10 °C rise in temperature, 95% CI: 0.59 to 0.84%) and C18:2t (0.15%, 95% CI: 0.01 to 0.23%) levels increased with heating, while levels of C18:3t decreased (−0.74%; 95% CI: −0.93 to −0.54%). There was no significant interaction between heating temperature and time for C18:1t. While significant interactions were observed for C18:2t and C18:3t at various temperature intervals, the interaction effect was quite modest (Figure 3).

## 4. Discussion

In our systematic review of 33 studies investigating the effect of heating edible oils on their TFA levels, we found that heating to temperatures most commonly used in cooking (≤200 °C) had minimal impact on TFA levels but heating to higher temperatures (>200 °C) could increase levels of TFA. The effect of heating on formation of individual types of TFA appeared to be largely consistent, leading to gradually increasing TFA levels with increasing temperature above 200 °C. Furthermore, levels of some TFA subtypes increased further with prolonged heating, especially at temperatures above 200 °C.

Our novel findings suggest that even at the temperature range >200 °C (where significant increase in TFA was observed), the magnitude of the increase appears to be relatively small. For instance, the expected increase in total TFA for an increase in temperature between 25 °C (room temperature) and 220 °C is 7.4% (after a median cooking time of 45 min of the included studies), whereas the level of TFA in partially hydrogenated vegetable oil, the major target of global TFA elimination, is typically around 25–40% [46]. Nonetheless, these findings affirm recommendations from several European countries that to minimise formation of harmful TFA, frying oil should not exceed common cooking temperatures (i.e., <200 °C) and support other public health recommendations that prolonged and/or repeated use of cooking oils should be avoided, which may be particularly relevant for some informal food sectors in low- and middle-income countries where such practices may be common [47,48,49,50]. However, there appears to be a lack of formal guidance on the repeated use of cooking oils at both an international and national level, with the Food Safety and Standards Authority of India being a notable exception [47,51]. There may be a need to support vendors in both the formal and informal food sectors to avoid the practice of reusing cooking oils, such as through targeted education programs or subsidised access to fresh oils and used oil waste removal.

Though we have found some evidence to suggest that under various heating conditions each of the subtypes of TFA assessed here and total TFA increase, C18:3t was found to be the TFA that most readily increased (i.e., even below 200 °C, it showed significant increase). This may indicate that the precursor fatty acid (C18:3, alpha-linolenic acid) is more susceptible to the effects of heating than other mono- and polyunsaturated fatty acids (i.e., oleic acid and linoleic acid for C18:1 and C18:2, respectively). This is a novel finding which suggests that as such, the avoidance of cooking oils that contain high levels of C18:3, such as various seed oils, in cooking methods reaching high temperatures may be a useful additional way to avoid the generation and consumption of TFA. It should be noted that 240 °C is above the smoking point of most cooking oils, i.e., the temperature at which oil starts to vaporise, which is typically undesirable for appearance, taste, and utility. Heating oils above their smoking point would not only impact the fat quality and lead to the generation of TFA but could also increase the levels of carcinogenic compounds [52]. Thus, cooking at temperatures much lower than 240 °C is important for avoidance of generation of such compounds in addition to TFA [43].

To our knowledge, this is the first systematic review that synthesised data on the relationship between TFA levels and cooking temperature and time. Strengths of this study include its relatively large sample size, standardisation of outcome measures that enabled pooling of data across studies, adjustment for individual study variation by use of mixed multilevel models with random intercepts for individual studies, adjustment for oil types, and analysis of changes in individual TFA isomers. Nevertheless, the results of this study should be interpreted in the context of some limitations. First, the varied results for different TFA isomers at temperatures above 240 °C reflect an overall paucity of data at this temperature range and should be interpreted with caution, although such cooking temperatures may have little real-world relevance as most common household cooking practices occur at temperatures between 160 and 200 °C. Second, pooled analyses could not be performed for individual oil types given relatively few data points. Rather, we adjusted for oil type in our overall mixed models, thereby retaining statistical power of our dataset. Third, given the relatively few data points investigating the change in TFA content of food items cooked in edible oils, we were unable to meaningfully pool data investigating change in TFA content of foods. This is an important consideration as the TFA level in frying materials and the frying oils have been shown to affect each other by the exchange of fats during cooking [43]. As such, more studies should evaluate how heating edible oils changes the TFA content of various foods cooked in those oils. Fourth, owing to the considerable heterogeneity in reported units between studies (e.g., % of total fatty acids, % of 16:0, % C18:1-9c, % of C18:2, % of C18:3, or mg/g of chicken leg meat), and despite consolidation of some categories, we were unable to pool data from studies that reported different units. Future studies should report consistent units, for example “% of total fatty acids”, to facilitate pooling across studies. Finally, for some TFA subtypes, the number of studies and experiments including controls at room temperature was few, reducing the statistical power of comparisons of mean TFA at different temperature intervals.

## 5. Conclusions

Our systematic review of the literature suggests that while heating edible oils to commonly used cooking temperatures has little effect on TFA generation, heating to higher temperatures and for a longer period of time can increase TFA levels. These findings provide further evidence that prolonged heating of edible oils to very high temperatures may be harmful and should be avoided to reduce dietary intake of TFA.

## Figures and Tables

**Figure 1 nutrients-14-01489-f001:**
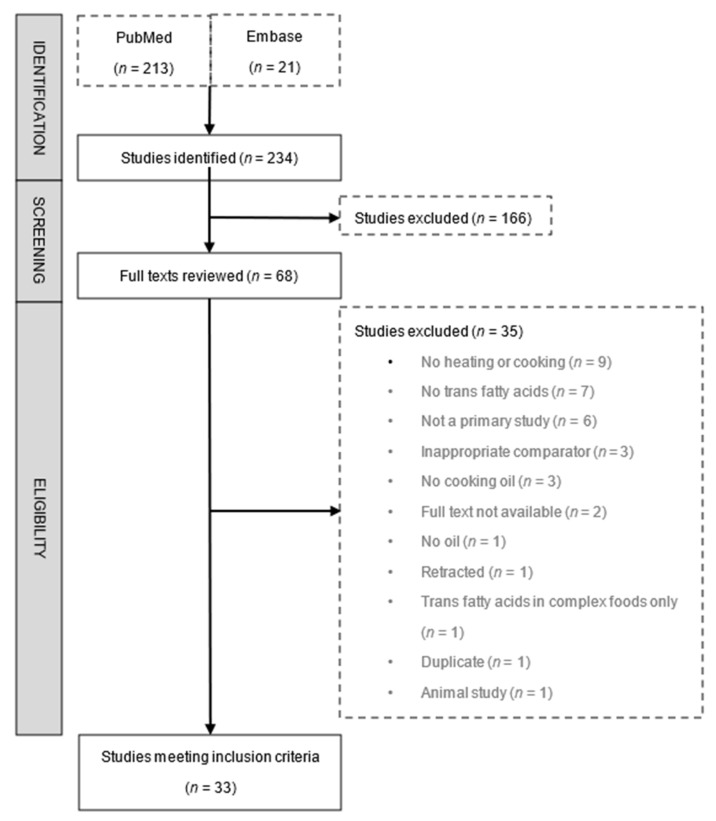
Flow diagram of study selection.

**Figure 2 nutrients-14-01489-f002:**
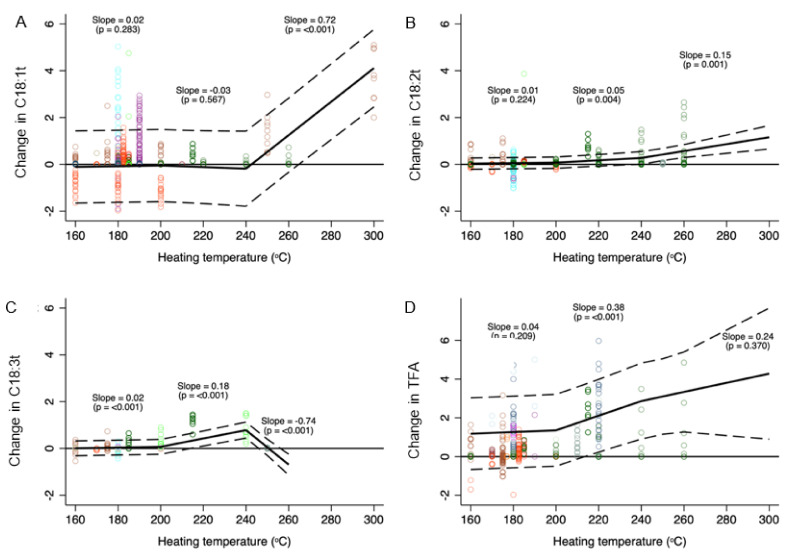
Change in C18:1t (**A**), C18:2t (**B**), C18:3t (**C**), and total TFA (**D**) (% of total fatty acids) content of cooking oil as a function of heating temperature. Data were fitted using a mixed multilevel linear regression model adjusted for heating time and oil type with random intercepts for studies and spline knots at 200 and 240 °C. Slopes represent change in TFA per 10 °C change in heating temperature within a particular spline range (<200 °C, 200–240 °C, >240 °C). Data point colours represent different studies.

**Figure 3 nutrients-14-01489-f003:**
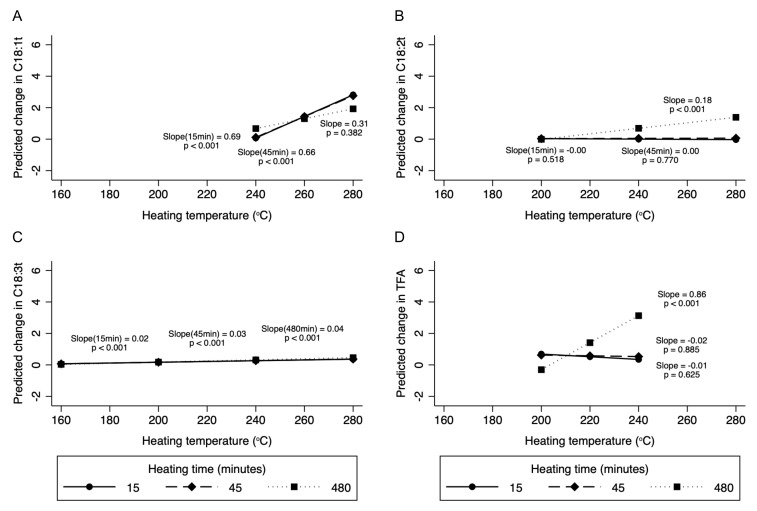
Margins plot demonstrating the interaction between heating temperature and heating time for various TFA. In subfigures (**A**), (**B**), and (**D**), the data label for 480 min is 0.31, 0.18, and 0.86, respectively, and in (**C**) it is 0.04. Data were fitted using mixed multilevel linear regression with intercepts for studies specified as random effects and splines at 200 and 240 °C. Slopes represent change in C18:1t (**A**), C18:2t (**B**), C18:3t (**C**), and total TFA (**D**) per 10 °C (% of total fatty acids) change in heating temperature. Margins plots were created for those temperature splines where there was a statistically significant change in TFA.

**Table 1 nutrients-14-01489-t001:** Levels of different *trans* fatty acid (TFA) (% of total fatty acids) as a function of heating temperature. Changes in TFA levels resulting from heating oils above room temperature as well as the difference in TFA levels in oils heated to high (>200 °C) versus usual cooking temperatures (≤200 °C) are depicted for each TFA studied.

		TFA Concentration (% of Total Fatty Acids)							
		Unheated		<200 °C		200–240 °C		>240 °C	
Fatty acid		*n* studies (*n* samples)	Median (IQR)/ Estimate (95% CI); *p*	*n* studies (*n* samples)	Median (IQR)/ Estimate (95% CI); *p*	*n* studies (*n* samples)	Median (IQR)/ Estimate (95% CI); *p*	*n* studies (*n* samples)	Median (IQR)/ Estimate (95% CI); *p*
16:1t	Median	0 (0)	-	4 (60)	0.02 (0.01; 0.03)	2 (30)	0.02 (0.01; 0.03)	1 (16)	0.02 (0.00; 0.02)
	Difference from usual cooking temperatures		-		Reference		0.00 (−0.00, 0.01); *p* = 0.62		−0.01 (−0.01, 0.00); *p* = 0.052
18:1t	Median	3 (14)	0.35 (0.06, 0.56)	13 (306)	0.24 (0.03, 1.49)	5 (79)	0.07 (0.01, 3.72)	3 (37)	1.08 (0.08, 3.20)
	Difference from unheated		Reference		0.71 (−1.11, 2.54); *p* = 0.45		0.61 (−1.26, 2.48); *p* = 0.52		2.06 (0.10, 4.02); *p* = 0.04
	Difference from usual cooking temperatures				Reference		−0.10 (−0.97, 0.77); *p* = 0.82		1.33 (0.07, 2.60); *p* = 0.039
18:2t	Median	1 (1)	0.13 (-)	10 (178)	0.31 (0.01, 0.50)	4 (69)	0.42 (0.03; 0.62)	1 (48)	0.48 (0.21; 0.98)
	Difference from unheated		Reference		0.33 (−0.51, 1.18); *p* = 0.45		0.38 (−0.47, 1.23); *p* = 0.38		0.76 (−0.10, 1.61); *p* = 0.084
	Difference from usual cooking temperatures				Reference		0.05 (−0.07, 0.18); *p* = 0.42		0.43 (0.28, 0.57); *p* < 0.001
18:3t	Median	2 (7)	0.01 (0.00, 0.20)	4 (56)	0.01 (0.00, 0.27)	3 (27)	0.01 (0.00, 0.30)	1 (12)	0.70 (0.53, 1.48)
	Difference from unheated		Reference		0.11 (−0.23, 0.45); *p* = 0.52		0.20 (−1.14, 0.53); *p* = 0.25		0.80 (0.44, 1.15); *p* < 0.001
	Difference from usual cooking temperatures				Reference		0.05 (−0.11, 0.21); *p* = 0.52		0.62 (0.37, 0.88); *p* < 0.001
Total TFA	Median	1 (1)	0.09 (-)	10 (117)	0.97 (0.62, 1.53)	5 (52)	1.42 (0.87; 3.70)	2 (17)	1.54 (0.97; 4.10)
	Difference from unheated		Reference		2.14 (−8.24, 12.52); *p* = 0.69		2.49 (−7.93, 12.90); *p* = 0.64		3.78 (−6.84, 14.40); *p* = 0.49
	Difference from regular cooking temperatures				Reference		0.34 (−1.42, 2.10); *p* = 0.70		1.64 (−1.22, 4.49); *p* = 0.26

## Data Availability

The data presented in this study are available on request from the corresponding author.

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
