# Peer review of "Influence of Heating during Cooking on Trans Fatty Acid Content of Edible Oils: A Systematic Review and Meta-Analysis"

_nutrients, 2022, doi:10.3390/nu14071489_

Round 1

Reviewer 1 Report

Dear authors,

I have reviewed the manuscript entitled "Influence of heating on trans fatty acid content of edible oils: A systematic review and meta-analysis". This is a clearly presented manuscript presenting the result of a systematic reviewing and meta-analysis of studies addressing the influence of temperature on trans-fatty acid formation in edible oils used in cooking procedures. I would like to offer a few minor comments which the authors may want to address in order tom improve this otherwise fine and interesting manuscript.

Comments:

  1. My main suggestion is to include in the title the wording "during cooking", or similar, in order to make it clear that the authors limited the study to effects of heating during cooking. The reason this refinement is desired is because heating during the refining of these edible oils has not been addressed and reviewed in this study. A substantial body of literature exists on the effect of temperature and heat exposure on trans fatty acid formation during refining. When I first read the title of this review, I assumed that the focus of the manuscript was going to be on that aspect of exposure of edible oils to heat, only to discover quickly that the focus was on cooking procedures. Although the title becomes a little longer, it could read as "Influence of heating during cooking on trans fatty acid content of edible oils: A systematic review and meta-analysis ".    Similarly, this or similar wording should be included in line 295 ("...on the relationship between TFA levels and heating temperature during cooking and time.").
  2. In my opinion - strictly taken - the observations made in this study did not lead to the conclusion that repeated heating also leads to increased TFA levels; the study addressed temperature dependency and a prolonged time exposure dependency, but has not addressed repeated heat exposures. Hence, it is necessary to change the words "repeated heating" to "prolonged heating" in the Abstract, line 25, and in the Conclusion section, line 322. In the abstract the authors write  "This" (line 24), which refers to "Our findings", line 22, and hence the conclusions refer to the findings of this study, and should therefore be restricted to the parameters that were addressed, which include prolonged exposure but not repeated exposure.
  3. Line 127 - analytical variability
  4. Line 178 - A question - is it truly possible to measure one-hundredth of a percentage changes in temperature? What is the precision in temperature measurements in the selected studies? Please double check, and decide if it is possible to draw the conclusion that indeed there is a statistically significant change for C18:3t levels in the below 200 degrees Celsius range 
  5. Fig 2C - The slope downwards from the break point at 240 degrees Celsius, is not accompanied by any individual datapoints reflecting the actual values for the corresponding studies. Please include, or explain their absence. On what is the negative slope based?
  6. Line 303 - It could also be of interest to indicate to the reader which temperature range you consider of relevance to real-life cooking situations (here, in the Discussion section, or in the Introduction)
  7. Line 313 - It is unclear what you mean with the statement ".., we were unable to pool a small proportion of our data". Please explain or rephrase
  8. Lines 325 - Please provide the names of the three titles
  9. The reference list needs some work: Some references do not have the journal names indicated with capital letters (eg 1 and 11). Some have each word of the title capitalized, others do not. Reference 8 contains unnecessary information about authors. Reference 9 does not have a correct journal abbreviation. Reference 42 does not have a journal name? 
  10. Please provide a website link for reference 51.

Reviewer 2 Report

The manuscript “Influence of heating on trans fatty acid content of edible oils: A systematic review and meta-analysis” discuss interesting subject. This is the first systematic review that synthesized data on the relationship between TFA levels and heating temperature and time. Comparable data were analyzed using mixed multilevel linear models taking into account individual study variation. Thirty-three studies encompassing 21 different oils were included in this review. The studies are comprehensively described and supported by the literature. They provide information that is a valuable addition to the existing knowledge. The manuscript deals with a very important topic related to food safety. However, it is not scientific or innovative. The conclusions presented at the end of the paper are widely known to researchers and food technologists. The factors influencing the formation / presence of trans isomers in food are obvious to the fat research community. Nevertheless, the manuscript is of great practical importance for the average consumer, for housewives or for street, small gastronomy. It would be good if the manuscript was sent to health institutions and that the guidelines for people contained guidelines on the conditions of use of oils and fats at home or in gastronomy. Consumption of trans fatty acids is associated with adverse health outcomes and considerable burden on morbidity and mortality globally.

The manuscript is prepared in a professional manner. The summary is sufficiently informative about the content of the manuscript and the conclusions of the research. The introduction is exhaustive. The aim of the work is clearly established. The methods are well described. The results obtained are well-discussed. The presentation of the results in tables and figures are appropriate and sufficient. The discussion with literature seems sparse and the scope of the meta-analysis performed is modest. The discussion is laconic and convinces about the validity of the research carried out.

List of literature - the record of the used literature is careless; in some  items, full names of journals are missing (sometimes full names are present, other times abbreviations), sometimes the names of journals are written in capital letters and sometimes in lower case; not all items have a DOI and some items have duplicate DOIs.
